UPDATE ARTICLE

# Cuticular collagens mediate cross-kingdom predator–prey interactions between trapping fungi and nematodes

**Han-Wen Chang[1,2], Hung-Che Lin[2], Ching-Ting Yang[2,3], Rebecca J. Tay[2], Dao-Ming Chang[4], Yi-Chung Tung[4], Yen-Ping Hsueh** [1,2,3]*

1 Molecular and Cell Biology, Taiwan International Graduate Program, Academia Sinica and Graduate Institute of Life Sciences, National Defense Medical Center, Taipei, Taiwan, 2 Institute of Molecular Biology, Academia Sinica, Taipei, Taiwan, 3 Max Planck Institute for Biology Tübingen, Tübingen, Germany 4 Research Center of Applied Sciences, Academia Sinica, Taipei, Taiwan

* ping.hsueh@tuebingen.mpg.de

The Editors encourage authors to publish research updates to this article type. Please follow the link in the citation below to view any related articles.

## Abstract

Adhesive interactions, mediated by specific molecular and structural mechanisms, are fundamental to host–pathogen and predator–prey relationships, driving evolutionary dynamics and ecological interactions. Here, we investigate the cellular and molecular basis of adhesion between the nematode *Caenorhabditis elegans* and its natural predator, the nematode-trapping fungus *Arthrobotrys oligospora*, which employs specialized adhesive nets to capture its prey. Using forward genetic screens, we identified *C. elegans* mutants that escape fungal traps and revealed the nuclear hormone receptor NHR-66 as a key regulator of fungal-nematode adhesion. Loss-of-function mutations in *nhr-66* conferred resistance to fungal trapping through the downregulation of a large subset of cuticular collagen genes. Restoring collagen gene expression in *nhr-66* mutants abolished the escape phenotype, highlighting the essential role of these structural proteins in fungal-nematode adhesion. Furthermore, sequence analysis of natural *C. elegans* populations revealed no obvious loss-of-function variants in *nhr-66*, suggesting selective pressures exist that balance adhesion-mediated predation risk with physiological robustness. We observed that loss of *nhr-66* function resulted in a trade-off of increased hypersensitivity to hypoosmotic stress and cuticular fragility. These findings underscore the pivotal role of structural proteins in shaping ecological interactions and the evolutionary arms race between predator and prey.

## Introduction

Predator–prey relationships are strong drivers of evolution, with the emergence of adaptations and counter-measures in both predators and their prey. This dynamic process frequently influences the expansion of gene families critical for interactions

**Data availability statement:** Most of the relevant data are within the paper and its Supporting information files. Differential gene expression analysis between N2 and *nhr-66* mutants files are available from the GEO database (GSM8967049–GSM8967054).

**Funding:** This work was supported by Academia Sinica Investigator Award AS-IA-111-L02 to Y.-P.H., Academia Sinica Postdoctoral Fellowship to R.J.T., and the Max Planck Society to Y.-P.H. The funders played no roles in the study design, data collection and analysis, decision to publish, or preparation of the manuscript.

**Competing interests:** The authors have declared that no competing interests exist.

**Abbreviations :** AFM, atomic force microscopy; AMPs, antimicrobial peptides; CaeNDR, *Caenorhabditis* Natural Diversity Resource; CGC, *Caenorhabditis* Genetics Center; EMS, ethyl methanesulfonate; ENU, N-Nitroso-N-ethylurea; GFOGER, glycine-phenylalanine-hydroxyproline-glycine-glutamate-arginine; GO, gene ontology; NG, nematode growth; NHR, nuclear hormone receptor; NTFs, nematode-trapping fungi; PCA, principal component analysis; SNP, single-nucleotide polymorphism; TEM, transmission electron microscopy; TEP, Trap Enriched Protein; WGS, whole-genome sequencing.

between species. For example, most plant and animal species possess a highly diverse array of antimicrobial peptides (AMPs), which play a key role in defending against bacterial and fungal infections [1]. The evolution of resistance to specific AMPs by pathogens, in turn, promotes the diversification and expansion of AMP gene families in animals [2]. This ongoing evolutionary arms race highlights the interconnectedness of life and the perpetual nature of adaptation.

The cross-kingdom predator–prey interactions between carnivorous fungi and nematodes, the most abundant animals on Earth, provide a fascinating example of coevolution in soil ecosystems [3]. Among these carnivorous fungi are nematode-trapping fungi (NTFs), a diverse group of over 300 soil-dwelling species that develop specialized hyphal structures like adhesive nets and constricting rings to trap prey under nutrient scarcity [4]. Our previous ecological studies in Taiwan revealed that NTFs and nematodes cohabit in over 60% of soil samples analyzed [5], underscoring the ubiquity of this predator–prey relationship. While predator-prey interactions play a vital role in driving ecological and evolutionary processes across the tree of life, the availability of a genetically tractable model system to study this fundamental question at the molecular and cellular level remains limited. As a result, the model nematode *Caenorhabditis elegans* and its natural predator, the NTF *Arthrobotrys oligospora,* provides a unique opportunity to genetically dissect the molecular details of predator–prey dynamics from both sides.

Adhesive interactions are physical and biochemical processes facilitated by specific molecules or structures that enable cells or organisms to bind to each other or to surfaces. These interactions are not only essential for the development and physiology of an organism but also play a pivotal role in interspecific relationships, such as host–pathogen and predator–prey interactions. Within the Orbiliaceae family, many NTFs have evolved specialized adhesive traps to capture nematodes [6]. *A. oligospora*, the most extensively studied species in this family, serves as a model for exploring the biology of carnivorous fungi and provides a powerful system for investigating the molecular mechanisms underlying cross-kingdom microbe-metazoan adhesion [7]. By studying how a predatory fungus traps a nematode prey, we can uncover parallels between predatory strategies and pathogenic mechanisms, shedding light on the pivotal role of adhesion in shaping ecological and evolutionary processes.

The adhesive nets of *A. oligospora*, crucial for binding to the nematode cuticle, consist of extensive layers of extracellular polymers, as demonstrated by transmission electron microscopy (TEM) and chemical characterization [8]. The reduced trapping efficiency of *A. oligospora* traps treated with pronase indicates that proteins are critical for nematode adhesion [8]. In agreement with this hypothesis, we have recently identified a gene family that is highly expressed in trap cells. These Trap Enriched Protein (TEP) genes are expanded in the genomes of NTFs specifically, suggesting they may be adaptive for nematode trapping [9]. While these findings highlight the importance of protein-mediated interactions, evidence suggests that carbohydrate binding may also be involved in the molecular mechanism by which *A. oligospora* targets nematodes [10]. Despite these findings, the precise molecular and genetic pathways mediating glycan binding in this adhesion process remain poorly understood.

Here, we aim to explore fungal-nematode adhesion with a focus from the side of nematodes. Using unbiased forward genetic screens, we isolated *C. elegans* mutants capable of escaping *A. oligospora* traps, and identified the transcription factor NHR-66 as a pivotal regulator of fungal-nematode adhesion. Transcriptomic and gene ontology (GO) analyses in the *nhr-66* mutant revealed significant downregulation in cuticular collagen gene family expression. Restoration of the expression of multiple collagen genes in the *nhr-66* mutant background rendered nematodes once again susceptible to *A. oligospora* trapping, highlighting the critical role of collagens in adhesion. Furthermore, we discovered that *nhr-66* exhibits low genetic variation, similar to that of essential genes in wild isolates of *C. elegans,* implying that loss of *nhr-66* function may lead to severe trade-offs that compromise nematode fitness. Indeed, we observed that *nhr-66* mutants are hyper-sensitive to hypoosmotic stress, likely due to a weakened cuticle structure that may be attributed to the loss of collagen expression. Together, these results suggest predatory fungi target abundant and diverse collagen proteins on the cuticle to ensure successful prey capture, which may have driven the expansion of the collagen gene family in nematode genomes.

## Results

### Forward genetics reveals *nhr-66* mutation confers nematode resistance to *A. oligospora* traps

To investigate the mechanism underlying the trapping interaction between *A. oligospora* and *C. elegans*, we performed a forward genetic screen to identify mutants capable of escaping fungal traps. We utilized ethyl methanesulfonate (EMS) or N-Nitroso-N-ethylurea (ENU) to induce mutations in wild-type *C. elegans* (N2). Subsequently, we screened for F2 progeny able to traverse through *A. oligospora* traps towards a food patch (OP50 bacteria) positioned opposite the fungal colony (Fig 1A). We selected five highly resistant mutants (S1 Video) for detailed investigation. These mutants demonstrated a striking ability to move freely across fungal traps, in stark contrast to wild-type *C. elegans*, which were immediately captured upon contact with *A. oligospora* traps. To quantify trap resistance in our five mutants, we measured their escape rate by exposing them to fungal traps and calculating the percentage of uncaptured nematodes relative to the total tested after 10 min. As expected, 100% of wild-type *C. elegans* were caught immediately by the traps and could not escape (0% escape rate). In contrast, the five screen mutants all showed a similarly high escape rate of approximately 40% (Fig 1B).

To pinpoint the causative mutations, we employed single-nucleotide polymorphism (SNP)-based genetic mapping. Briefly, we crossed mutants to Hawaiian (CB4856) and genotyped the F2 progeny that escaped fungal trapping for strain-specific SNPs. SNP mapping revealed a strong correlation between the resistance phenotype and chromosome IV (Fig 1C; see S1 Data for details) [11]. We then performed whole-genome sequencing (WGS) of the five mapped F2 lines, with each derived from a unique P0 isolated during initial screening. WGS analysis identified six independent mutations, including loss-of-function alleles, that affected all isoforms of the gene *nhr-66* (Fig 1D). To confirm the role of *nhr-66* in fungal-nematode interactions, we added a STOP-IN cassette [12] at the *nhr-66* locus *(yph413)* in the N2 background using CRISPR/Cas9. We then phenotyped the STOP-IN strain for resistance to *A. oligospora* traps. Consistent with the *nhr-66* mutants from our screen (*yph406-408, 410,* and *412*), the CRISPR-generated *nhr-66* STOP-IN mutant displayed resistance to fungal trapping (Fig 1B). Moreover, rescue using a fosmid vector containing the full-length *nhr-66* genomic locus successfully reversed the resistance phenotype in both the genetic screen mutants and the STOP-IN mutant (Fig 1B), strongly supporting a critical role for *nhr-66* in nematode trapping by *A. oligospora*.

### NHR-66 mediates fungal adhesion through expression in hypodermal and seam cells

Previous studies have identified NHR-66 as a transcription factor expressed in the hypodermis and seam cells [13]. To confirm *nhr-66* expression, we created a GFP reporter line using 2 kb of gDNA sequence upstream of isoform a of *nhr-66* as the promoter. The *Pnhr-66::GFP* reporter indeed revealed that *nhr-66* is prominently expressed in the hypodermis and seam cells, which are the key tissues for cuticle synthesis [14]. Additionally, *nhr-66* expression was detected in the intestine, head muscle cells, and head and tail neurons. This broad expression pattern suggests that NHR-66 may have multiple roles in different tissues (S1 Fig).

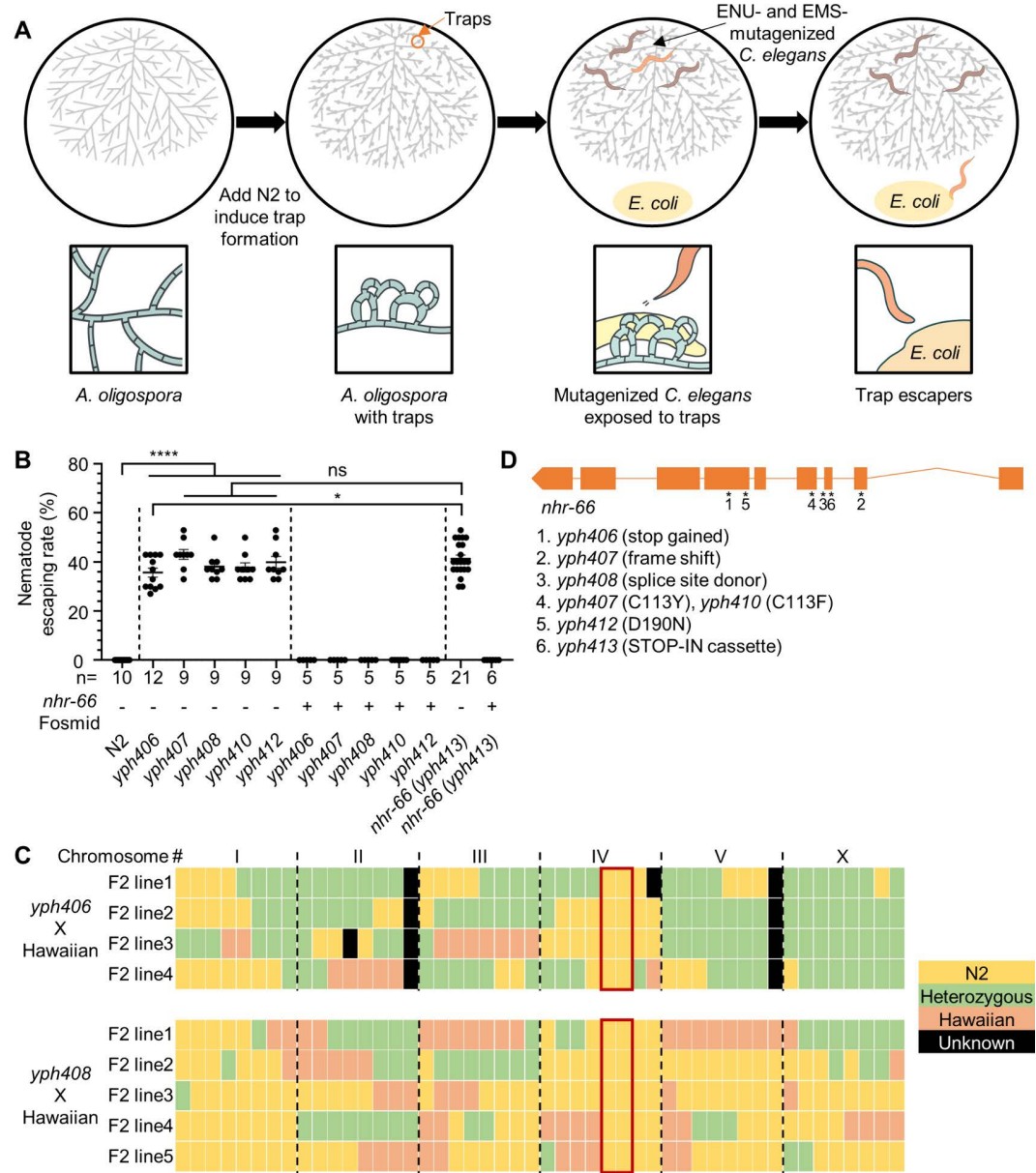

**Fig 1. Mutations in *nhr-66* promote *Caenorhabditis elegans* escape from the predatory fungus *A. oligospora*.** (A) Schematic of the forward genetic screen for trap-resistant *C. elegans*. (B) Quantification of nematode escaping rates in strains obtained from forward genetic screens (*yph406-408, 410,* and *412*), CRISPR STOP-IN (*yph413*), and fosmid-rescued strains (yphXXX alleles correspond to *nhr-66* variants shown in Fig 1D) (mean ± SEM; n is shown below the x-axis; two-tailed unpaired Student *t* test) (See S1 Video for more). (C) SNP mapping of two of the screen mutants. Trap-resistant mutants were crossed with Hawaiian strain CB4856, and the F2 progeny from each mutant line which exhibited trap resistance were analyzed for eight SNP markers on each chromosome. Regions containing only N2 SNPs (indicated by the red rectangle) were assumed to contain the causative mutation (See S1 Data for more). (D) Genetic mapping and whole-genome sequencing of trap-resistant mutants reveal causative mutations (marked by asterisks) in the gene *nhr-66*. Boxes and lines represent exons and introns, respectively. The data underlying this figure can be found in S1 Data.

To determine the site of action for NHR-66 in mediating fungal adhesive interactions, we reintroduced wild-type *nhr-66* cDNA driven by tissue-specific promoters into the *nhr-66(yph413)* mutant strain and assessed the ability of these rescued lines to escape *A. oligospora* traps (Fig 2A). Promoters that target *nhr-66* expression in various tissues were used: *unc-31* for pan-neurons, *ges-1* for intestine, *myo-3* for body wall muscle, *nhr-73* for seam cells, and *rol-6* for hypodermis [15–17]. We found that expressing *nhr-66* in neurons, intestine, or body wall muscle did not restore the adhesive interaction to *A. oligospora* (Fig 2A). However, expression of *nhr-66* in seam cells and hypodermal tissue successfully rendered the nematodes susceptible to fungal trapping, indicating that these tissues are the site of action for NHR-66. Given that *nhr-66* expression in the hypodermal and seam cells is essential for mediating adhesive interactions to *A. oligospora* traps, we hypothesized that NHR-66 likely plays a key role in regulating the physical and chemical properties of the cuticle in *C. elegans*.

### Mutation in *nhr-66* did not alter the macroscopic structure of the cuticle in *C. elegans*

We next asked whether loss of *nhr-66* alters the macroscopic structure of the cuticle and thereby affects the adhesive interaction with fungal traps. To test this, we employed atomic force microscopy (AFM) to examine the cuticle structure of wild-type (N2) and *nhr-66(-)* animals. The nematode cuticle features two primary surface structures: annuli and alae [18]. Upon AFM examination, no significant differences were detected in these surface structures between the wildtype and *nhr-66* mutant (Fig 2B and 2C). Additionally, we investigated whether cuticle thickness is affected by loss of *nhr-66*. Using TEM to analyze cross-sections of wildtype and *nhr-66* mutants, we measured the total cuticle thickness and the thickness of distinct cuticle layers, including the basal, medial, and cortical sections. The results showed no significant differences in cuticle thickness between N2 and *nhr-66(yph413)* mutants, although strut width was slightly narrower in the *nhr-66* mutant (Fig 2D). Overall, these findings revealed that the macroscopic structure of the cuticle surface in *nhr-66* mutants is similar to wildtype.

### NHR-66 is a master regulator that regulates the expression of the collagen gene family in *C. elegans*

Since NHR-66 is a predicted nuclear hormone receptor (NHR), we performed RNA-seq analysis on both wildtype and *nhr-66* STOP-IN mutants to uncover the potential targets regulated by NHR-66. Principal component analysis (PCA) showed distinct clustering of the N2 and *nhr-66* mutant replicates, indicating substantial transcriptional differences between the two genotypes (Fig 3A). In the *nhr-66* mutant, expression of more than 1,200 genes was significantly downregulated (>4-fold), whereas a comparatively small number of genes (168) were upregulated compared to wildtype, suggesting that NHR-66 mainly functions as a transcriptional activator (Fig 3B; see S3 Data for details). To further investigate the genes regulated by NHR-66, we performed GO enrichment analysis of the genes that were down-regulated in *nhr-66(-)*, which revealed a significant enrichment of genes involved in 'structural constituent of cuticle' (GO:0042302) and 'collagen trimer' (GO:0005581), suggesting that NHR-66 plays a critical role in regulating the expression of cuticular collagens (Fig 3C; see S3 Data for details) [19,20]. Indeed, our analyses revealed that over 40% (73/173) of genes in the cuticular collagen gene family in the *C. elegans* genome were downregulated in the *nhr-66* mutant. These results suggest that collagen proteins might be critical for fungal-nematode adhesive interactions. To test this hypothesis, we examined the adhesion to *A. oligospora* of 20 collagen mutants, such as *col-36* and *dpy-7*, and found that none of them showed resistance to trapping by *A. oligospora* (Fig 3D). These results suggest functional redundancy of these collagen proteins in mediating the adhesive interactions between the nematode cuticle and fungal traps.

To determine whether collagen proteins are sufficient to complement the fungal-nematode adhesion defects in *nhr-66* mutants, we chose five collagen genes that were strongly down-regulated in *nhr-66(-)*, share high sequence homology across the *Caenorhabditis* genus, and represent three of the five major cuticle clusters [21] (Cluster B to D) (Fig 3E) for over-expression using the hypodermis-specific *rol-6* promoter. We found that the overexpression of any single collagen, regardless of cluster, in the *nhr-66* mutant significantly decreased its nematode escaping rate. As more collagens were rescued in

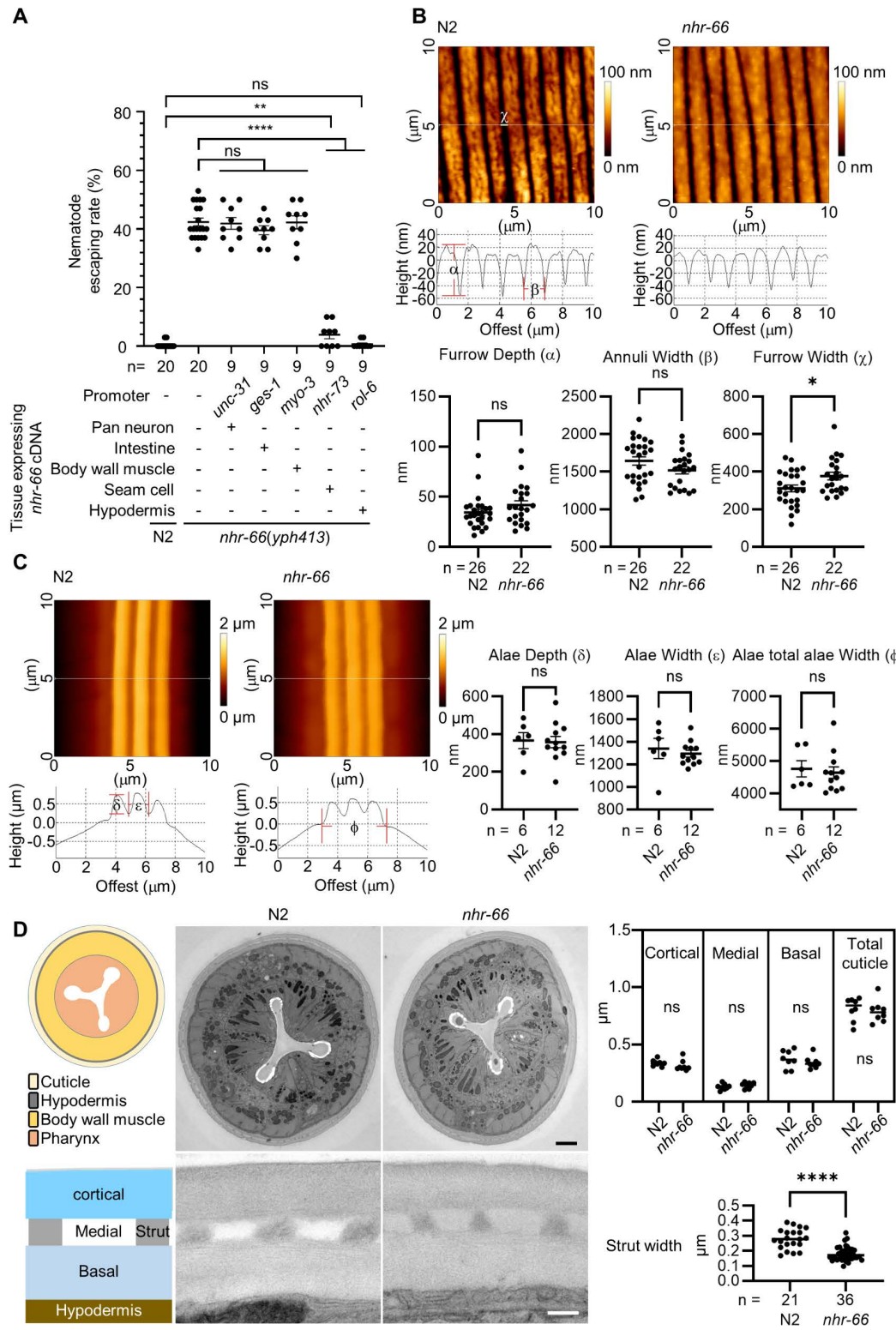

**Fig 2. NHR-66 promotes adhesion through cuticle-secreting tissues without altering cuticle structure. (A)** Quantification of nematode escaping rates in wild-type *Caenorhabditis elegans* (N2), the *nhr-66(yph413)* mutant, and tissue-specific rescue lines expressing *nhr-66* cDNA under various

promoters (mean ± SEM; n is shown below the x-axis; two-tailed unpaired Student $t$ test). **(B, C)** Atomic Force Microscopy (AFM) topography images (top), quantification and their line profiles (bottom) of adult cuticle annuli, furrows (B) and alae (C) for both wildtype and *nhr-66(yph413)* mutant. Data are presented as Mean ± SEM (n is shown below the x-axis; two-tailed unpaired Student $t$ test). **(D)** TEM images and quantification of cuticle (top) and strut width (bottom) from wild-type (N2) and *nhr-66(yph413)* animals (scale bar: white, 0.2 μm; black, 2 μm). Data are presented as Mean ± SEM (n is shown below the x-axis; two-tailed unpaired Student $t$ test). The data underlying this figure can be found in S2 Data.

the *nhr-66* mutant, escape rates further declined, with overexpression of all five nearly eliminating resistance (Fig 3F). These findings suggest that collagens are critical and sufficient for the adhesive interaction between *C. elegans* and *A. oligospora*.

To confirm that NHR-66 regulates the expression of collagens, we chose one collagen downregulated in *nhr-66* RNA-seq, *col-14*, and placed 2 kb of its promoter upstream of GFP. We then introduced the P*col-14*::GFP reporter into N2, the *nhr-66* STOP-IN mutant, and the *nhr-66* STOP-IN mutant rescued with hypodermal-driven (*rol-6* promoter) *nhr-66*. Fluorescence imaging showed reduced GFP expression in the *nhr-66* mutant compared to N2, and the GFP expression was restored in the *nhr-66* mutant rescued with the hypodermal promoter (Fig 3G). These results suggest that NHR-66 regulates the expression of cuticular collagen genes.

### NHR-66 is required for fungal-nematode adhesion across multiple species of nematode-trapping fungi

To determine whether NHR-66 plays an evolutionarily conserved role in nematode-fungal dynamics, we investigated whether *nhr-66* is resistant to trapping by various species of NTFs.

We tested 10 additional *Arthrobotrys* species, all employing adhesive traps, for their ability to capture wild-type *C. elegans* and *nhr-66* mutants (Fig 4A). We found that compared to wild-type animals, *nhr-66* mutants showed higher escaping rates across all species of NTFs, suggesting that collagen-mediated adhesive interaction with fungal trap is evolutionarily conserved. Additionally, we observed that *nhr-66* is less resistant to *A. musiformis*, suggesting that additional or alternative mechanisms of fungal-nematode adhesion likely have evolved in certain species of predatory fungi (Fig 4A).

### Natural variants of *nhr-66* in *C. elegans* wild isolates maintain normal fungal-nematode adhesion

Given that loss of *nhr-66* allows nematodes to escape fungal trapping under laboratory conditions, we next explored whether mutations in *nhr-66* might exist in wild *C. elegans* populations that might allow nematodes to escape predation. We examined the genomes of *C. elegans* wild isolates available from the *Caenorhabditis* Natural Diversity Resource (CaeNDR) [22] database, and identified only ~9% (138/1524) of wild isolates contain missense variants in *nhr-66*. Additionally, we identified a single in-frame deletion/start-loss mutation that affects only isoforms b—which is not expressed in the hypodermis—and j [23]. These findings suggest that *nhr-66* is highly conserved and subject to strong purifying selection in natural populations of *C. elegans* (Fig 5A; see S1 Table for details). We chose seven strains encoding one or more missense variants in distinct domains of the NHR-66 protein to test their resistance to *A. oligospora,* and found that none of these variants conferred trap resistance (Fig 5B). These findings suggest that in *C. elegans*, while loss of *nhr-66* function may provide a significant survival advantage against NTF predation, it is likely accompanied by substantial fitness trade-offs.

### Loss of NHR-66 enhances susceptibility to hypoosmotic stress

What could be the trade-offs associated with *nhr-66* loss-of-function? We observed that *nhr-66* mutants developed similarly to wildtype, reaching L4 and adulthood at 56 and 72 hours, respectively, after egg laying at 20 °C (S2 Fig). Furthermore, we observed no significant differences in dauer formation between N2 and *nhr-66* STOP-IN mutants when treated with extracted dauer pheromones [24] (S3 Fig). Next, we examined the brood sizes in *nhr-66* mutants and found them to be larger than those of N2 (S4 Fig), suggesting that loss of *nhr-66* may actually confer a reproductive advantage and fertility does not represent a trade-off. Further analysis of mating behavior [25,26] revealed higher mating efficiency in *nhr-66* STOP-IN mutants males compared to N2 (S5 Fig).

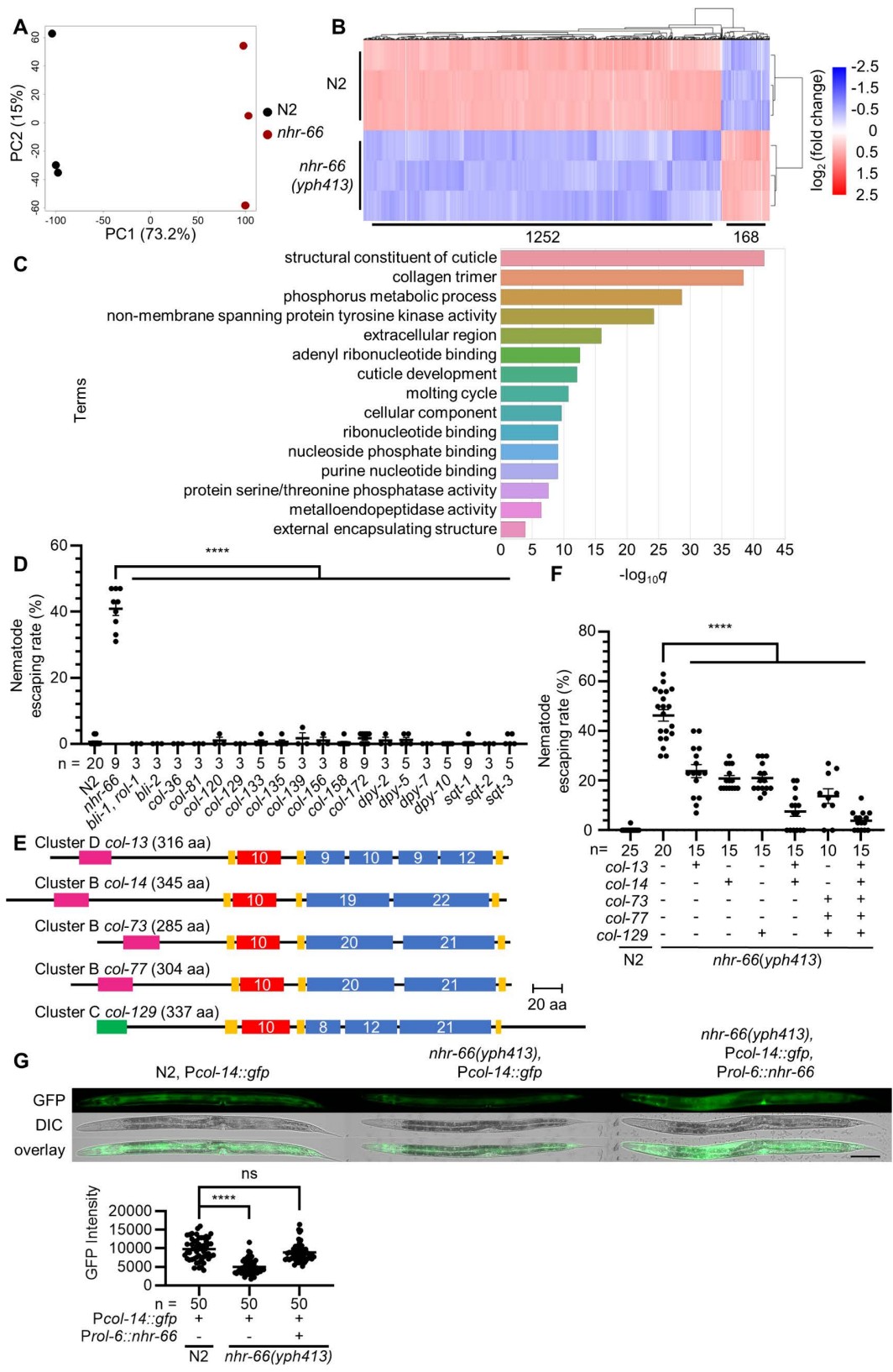

**Fig 3. NHR-66 regulates the expression of the collagen gene family in *Caenorhabditis elegans*. (A)** Principal component analysis of transcript expression from wildtype (N2) and *nhr-66(yph413)* mutants across three biological replicates. **(B)** Heatmap comparison of differentially expressed genes (≥ 4-fold change) between N2 and *nhr-66(yph413)* mutants. The numerical labels beneath the heatmap indicate the number of differentially expressed genes between N2 and the *nhr-66(yph413)* mutant. (See S3 Data for more). **(C)** Gene ontology enrichment analysis results for genes down-regulated more than 4-fold in the *nhr-66(yph413)* mutant. (See S3 Data for more). **(D)** Quantification of nematode escaping rates in wild-type *C. elegans* (N2), the *nhr-66(yph413)* mutant, and collagen mutants (mean±SEM, n is shown below the x-axis, two-tailed unpaired Student *t* test). **(E)** Protein structure of collagens used in rescue experiments. Collagens are classified into clusters based on interruptions in their main collagenous domain (blue boxes), with numbers in the boxes indicating the count of Gly-X-Y repeats. All cuticular collagens typically contain cysteine-rich regions (yellow), an N-terminal helical Gly-X-Y repeat (red box), a transmembrane region (pink) or predicted signal peptides (green). **(F)** Quantification of nematode escaping rates in collagen-rescued *nhr-66(yph413)* mutants (mean±SEM, *n* is shown below the x-axis, two-tailed unpaired Student *t* test). **(G)** Fluorescence images and quantification of P*col-14*::GFP reporter expression in N2, the *nhr-66(yph413)* mutant, and rescued strains (scale bar, 100 μm). Data are presented as the Mean±SEM (n is shown below the x-axis, two-tailed unpaired Student *t* test). The data underlying this figure can be found in S3 Data.

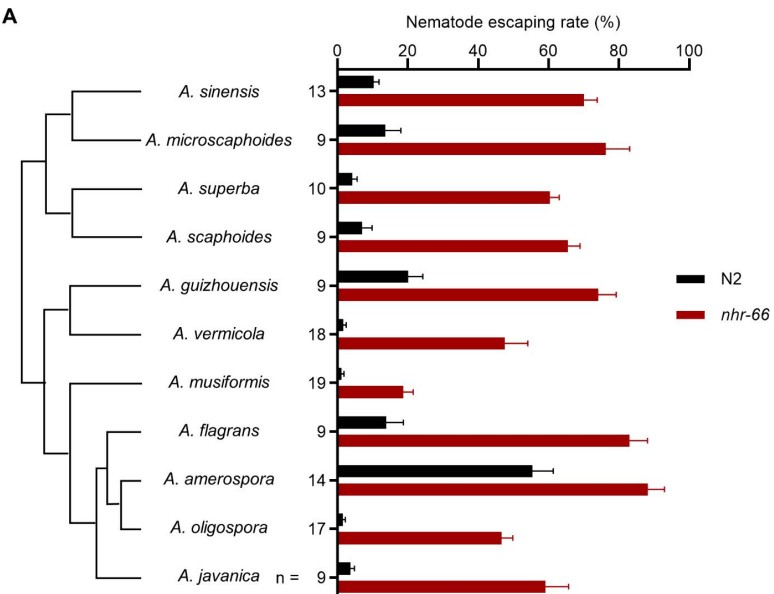

**Fig 4. The *nhr-66* mutant is resistant to multiple species of nematode-trapping fungi of the genus *Arthrobotrys*. (A)** Nematode escape rates of *nhr-66* mutants exposed to various species of nematode-trapping fungi (NTF phylogeny shown on the left). The phylogeny is based on single-copy orthologs. Data are presented as Mean±SEM (*n* is shown on the left the y-axis, two-tailed unpaired Student *t* test, *P*-values comparing WT and *nhr-66* mutant against each NTF strain are <0.001 with the exact *p*-values provided in the source data). The data underlying this figure can be found in S4 Data.

Given that *nhr-66* mutants exhibit lower collagen expression than N2, we hypothesized that *nhr-66* loss may compromise cuticle integrity. To test this, we performed hypoosmotic stress assays by immersing adult nematodes in deionized water and measuring the percentage that ruptured. Mutants in *nhr-66* showed a significantly higher explosion rate than N2, with ~60% of the animals exploded in water in 10 min, indicating a weakened cuticle barrier (Fig 5C and S2 Video). This result demonstrates that *nhr-66* mutations are unlikely to be maintained in *C. elegans* populations in natural environments. To evaluate the impact of *nhr-66* mutations on survival in natural environments, we conducted a competition assay between wild-type and *nhr-66* mutants. Equal numbers of GFP-labeled wild-type and *nhr-66* nematodes were introduced into soil, and populations were harvested after 1–3 weeks. Wild-type nematodes outcompeted *nhr-66* mutants, indicating reduced fitness of *nhr-66* mutants in a soil environment (Fig 5D). Collectively, these findings suggest that loss of *nhr-66* reduces capture by NTFs at the cost of compromised cuticle integrity and survival, highlighting a trade-off in natural conditions.

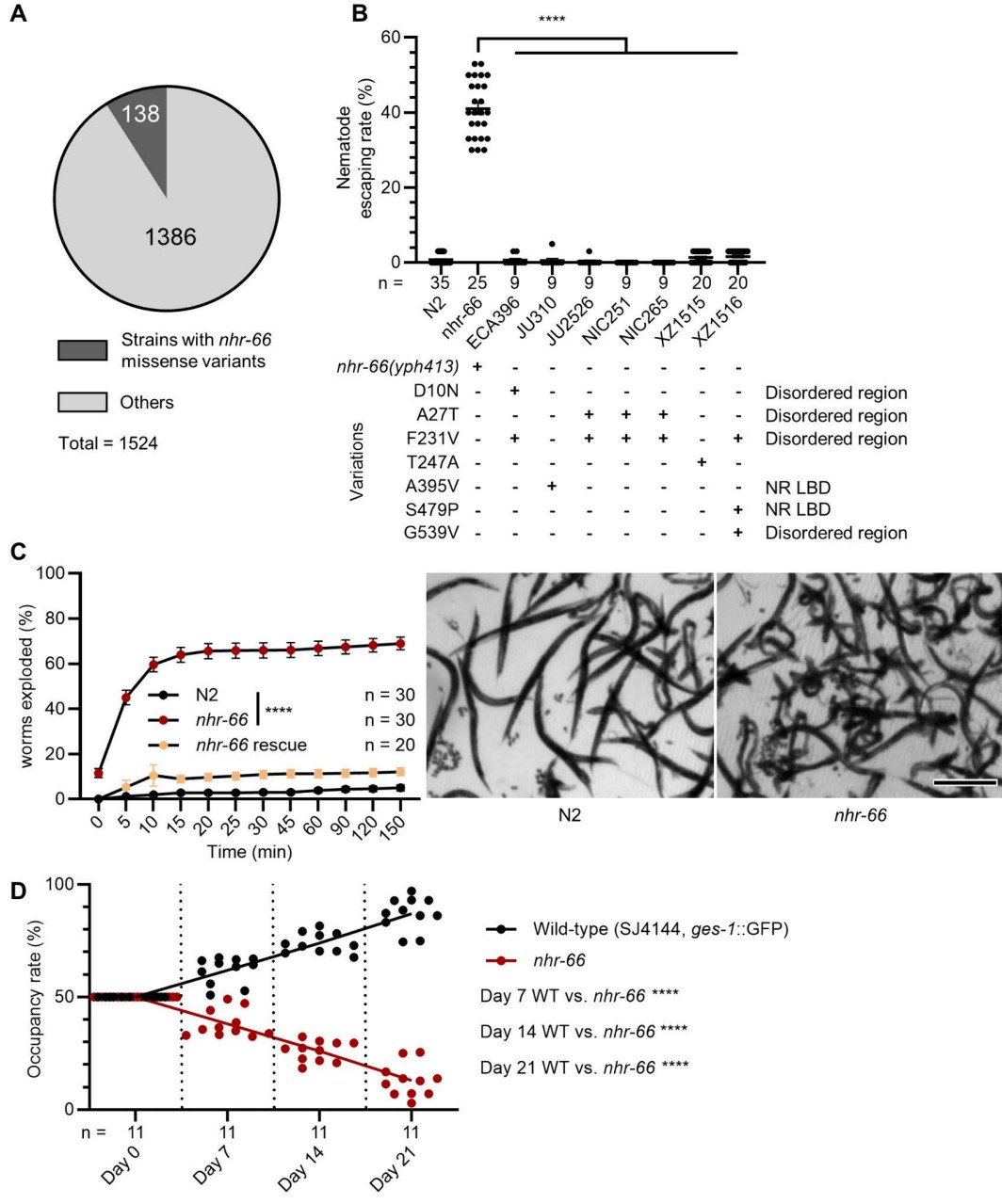

**Fig 5. nhr-66 is critical for the fitness and survival of *Caenorhabditis elegans* in natural environments. (A)** A pie chart showing the number of wild isolates bearing *nhr-66* missense variants out of the total number of isolates characterized on CaeNDR. (See S4 Table for more). **(B)** Nematode escaping rates in wildtype, *nhr-66(yph413)* mutants, and *nhr-66* natural variants (mean±SEM, n is shown below the x-axis; two-tailed unpaired Student *t* test). **(C)** Percentage of nematodes that exploded under hypoosmotic stress (mean±SEM, n is shown in the figure; two-tailed unpaired Student *t* test). Right, images of nematodes exposure to deionized water for 15 min (scale bar, 500 μm) (See S2 Video for more). **(D)** Occupancy rates of wildtype and the *nhr-66(yph413)* mutant in a soil competition assay at 20 °C (*n* is shown below the x-axis, two-tailed unpaired Student *t* test). The data underlying this figure can be found in S5 Data.

# Discussion

Predator–prey interactions drive fundamental ecological processes, including population dynamics, species distributions, and evolution. These interactions often involve specialized adaptations in both predator and prey. One critical predator adaptation is the ability to adhere to prey during the capture process, particularly in non-motile predators such as carnivorous plants and fungi. NTF utilize specialized adhesive structures—such as nets, knobs, and columns—to ensnare nematodes, making adhesion essential for their predatory efficiency and, consequently, growth and survival. Despite its significance, the molecular mechanisms behind NTF-nematode adhesion remain unclear. In this study, we employed forward genetic screens to investigate *C. elegans* factors important for fungal adhesion. Our study uncovered NHR-66 as a key regulator of nematode-fungal adhesive interactions, with *nhr-66* mutants exhibiting pronounced resistance to fungal traps. Our findings reveal, for the first time, a genetic basis for nematode resistance to fungal predation, shedding light on the interplay between nematode cuticle composition and fungal adhesion mechanisms.

Our forward genetic screen isolated five strong mutants. Interestingly, SNP mapping and WGS showed that the resistance phenotype we observed in each of these five strains could be explained by mutations in the single gene *nhr-66*, suggesting that our screen had reached or was close to saturation. Furthermore, the fact that *nhr-66* is a transcription factor, also indicated that only when a master regulator was mutated, a strong phenotype can then be revealed. Given its role as a transcription factor, we hypothesized that the downstream targets of NHR-66 affect trap resistance.

Hormone receptors are known to mediate systemic responses to environmental stimuli. Several NHRs, including NHR-23 and NHR-41, regulate the expression of cuticular collagen genes during development and molting [28,29]. For example, dynamic remodeling of the nematode cuticle by NHRs in the dauer stage can alter collagen expression for enhanced surface impermeability and stress resistance [30,31].

Previous studies have shown NHR-66 interacts with NHR-49 to transcriptionally repress genes in lipid metabolism [27,32]. To see if this previously described function of NHR-66 was crucial for trapping, we tested the escaping rate in a *nhr-49* mutant. The *nhr-49* mutant did not exhibit trap resistance, supporting a lipid-independent role for *nhr-66* in regulating cuticle properties (S6 Fig). In addition, our transcriptomic data suggest that, in contrast to previous reports of NHR-66 as a repressor, NHR-66 can positively regulate gene expression, particularly of cuticular collagens. Indeed, we discovered over 1,200 genes downregulated in *nhr-66* mutants. Thus, we propose that *A. oligospora* targets nematodes through the engagement of multiple molecular targets on the *C. elegans* cuticle and that these targets act redundantly to ensure adhesion between the fungus and nematode. In support of this model, single collagen mutants did not exhibit resistance to fungal traps (Fig 3D).

Approximately 40% of collagens were downregulated in the *nhr-66(yph413)* mutant, and overexpressing just a few of these collagens was sufficient to render the mutant animals susceptible to fungal traps (Fig 3F). The downregulation of collagens in the *nhr-66* mutant may explain the thinner struts we observed in these animals (Fig 2D). Specifically, BLI-1 and BLI-2, two collagens reported to localize at cuticle struts [33], are significantly downregulated in *nhr-66(yph413)* mutants. Their depletion may contribute to reduced strut thickness, potentially weakening the cuticle and increasing its vulnerability to hypoosmotic stress.

How does *A. oligospora* target collagens? One potential mechanism involves the recognition of collagen motifs. Collagens are characterized by a repetitive Gly–X–Y sequence, where X and Y are frequently proline and hydroxyproline, respectively; these polypeptide chains associate to form a right-handed triple helix that constitutes the hallmark structure of collagen. In vertebrate collagens, specific sequence motifs—such as the glycine-phenylalanine-hydroxyproline-glycine-glutamate-arginine (GFOGER) sequence—are recognized by integrin receptors (e.g., α1β1 and α2β1), facilitating robust cell–matrix interactions [34]. Although the exact same motifs do not appear in *C. elegans* collagens, similar recognition motifs may exist in nematode collagens. While nematode collagens differ structurally from their vertebrate counterparts, such as exhibiting reduced molecular size and a higher frequency of disrupted Gly-X-Y repeats [21], fungal adhesion proteins may nevertheless exploit integrin-like or other proteins to recognize nematode collagens.

Alternatively, *A. oligospora* may recognize nematode-specific modifications on collagens rather than a certain primary sequence. Many collagens in non-nematode animals, such as type I and type IV in mammals, contain hydroxylysine residues that are glycosylated with sugars such as galactose or glucosylgalactose. This process enhances collagen stability and function in tissues like skin and bones [35]. In contrast, nematode cuticular collagens, which form the exoskeleton, do not have significant amounts of hydroxylysine and thus may lack this type of glycosylation. While typical glycosylation in vertebrates involves addition of a Glc-Gal disaccharide, glycosylation in *C. elegans* can mean the addition of a complex N-glycan [36]. Although the precise glycosylation profile of nematode collagens remains unclear, these unique glycan modifications may serve as ligands for fungal lectins or other carbohydrate-binding proteins, mirroring glycosylation-dependent adhesion mechanisms described in other biological systems [10].

It is also conceivable that both glycan modifications and the collagen polypeptides themselves cooperate to facilitate fungal adhesion. Such a dual-targeting model—wherein fungal proteins recognize specific sugar moieties while simultaneously engaging with collagen peptide motifs—could enhance binding specificity and strength.

While our findings indicate that NHR-66 regulates fungal adhesion primarily through control of collagen expression, additional targets may contribute to nematode susceptibility. In addition to the collagen gene family, our RNA-seq analysis also identified several candidate gene families that might play a role in fungal-nematode interactions. For instance, C-type lectins—a diverse class of carbohydrate-binding proteins implicated in immune recognition and host–pathogen interactions—may facilitate nematode attachment to *A. oligospora* traps, as approximately 14% (38 of 264) of these genes are differentially expressed following fungal exposure. Further exploration of these C-type lectins could reveal additional molecular pathways involved in nematode–NTF adhesion.

From the perspective of NTF, adhesion between predator and prey is a critical step in successful predation. In *A. oligospora*, genes encoding TEPs are markedly upregulated upon nematode exposure and exhibit selective expansion across NTFs [9]. Specifically, TEP1 has been experimentally confirmed to play a crucial role in the adhesiveness of fungal traps. These proteins may function as versatile adhesion effectors capable of recognizing both glycosylation patterns and peptide sequences on the nematode cuticle. Further elucidation of the molecular interactions between fungal adhesion proteins and nematode collagens will be critical to understanding the mechanistic basis of fungal predation.

The nematode-NTF relationship exemplifies the Red Queen Effect, where predator and prey continuously evolve to counteract each other's adaptations. The expansion and utilization of new nematode collagen and C-type lectin genes may represent a defense response to escape fungal predation. Interestingly, while resistant to capture by most NTFs, *nhr-66* mutants remain susceptible to *Arthrobotrys musiformis*. This finding suggests that certain species of NTF may have also evolved additional adhesion mechanisms, perhaps as a countermeasure to adaptive evolution by nematodes.

Our analysis of the CaeNDR database provides further insight into the natural variation of *nhr-66* in *C. elegans* wild isolates. The predominance of silent and non-coding variants, along with the rarity of missense mutations and a single in-frame deletion/start-loss mutation affecting only *nhr-66* isoforms b and j, suggests that complete loss-of-function mutations are under strong negative selection (S1 Table). From an evolutionary perspective, the scarcity of *nhr-66* loss-of-function alleles in natural populations highlights a potential trade-off between fungal resistance and physiological stability. While resistance to fungal adhesion is beneficial, NHR-66 function appears essential for nematode fitness, particularly under hypoosmotic stress which nematodes may experience in wet environments, emphasizing the evolutionary balance between flight (adaptations for escaping predators) and forage (daily life). In contrast, examination of variants in cuticular collagen genes revealed that several collagens, such as *col-19*, *col-139*, and *sqt-2*, harbor high-impact mutations including start codon loss, frameshifts, and splice region variants. These observations suggest that, unlike *nhr-66*, certain collagen genes are able to tolerate greater variation at the protein level. This is likely due to *nhr-66* function as a master regulator with pleiotropic roles, whereas individual collagens primarily serve structural functions, such as protection and support. Moreover, the high degree of redundancy among collagen genes may buffer the effects of deleterious mutations, whereas no single factor appears capable of fully compensating for the diverse functions of *nhr-66*.

Understanding the mechanism of adhesion has broad implications beyond nematode-fungus interactions. Elucidating fungal adhesion pathways could inform the development of antifungal therapeutics, while insights into the structural and biochemical properties of the nematode cuticle may inspire novel biomimetic adhesives. Moreover, investigating nematode collagens offers valuable perspectives on extracellular matrix biology and wound healing, due to their structural similarities with vertebrate collagens [37]. Targeting the surface properties of nematodes may also facilitate the development of interventions against parasitic species, while optimizing fungal adhesion could enhance the efficacy of NTFs as biological control agents in agriculture. This work uncovering NHR-*66* as a key regulator of fungal adhesion enriches our understanding of host–pathogen/predator–prey interactions and provides a framework for understanding the genetic and molecular determinants of the evolutionary arms race between the most abundant animal group and their natural predators.

## Materials and methods

### Nematode and fungal strains

Nematode strains, including both wild-type and mutant variants, were maintained at room temperature on nematode growth (NG) medium agar plates supplemented with *Escherichia coli* OP50 as the bacterial food source. The wild-type strain *C. elegans* Bristol N2 was utilized as the reference strain, except where noted. Mutant strains were procured from the *Caenorhabditis* Genetics Center (CGC) or created in-house using CRISPR-Cas9 gene-editing technology (STOP-IN), as previously described [12]. Transgenic nematodes were generated using standard microinjection techniques to introduce plasmid constructs into the germline [38]. The NTF *Arthrobotrys oligospora* strain TWF154 served as the wild-type strain in this study and was used in most fungal assays, except experiments assessing the role of *nhr-66(-)* in resistance against different NTFs [5]. For all assays requiring trap induction, the fungi were cultivated on low-nutrient media (LNM) (2% agar, 1.66 mM $MgSO_4$, 5.4 µM $ZnSO_4$, 2.6 µM $MnSO_4$, 18.5 µM $FeCl_3$, 13.4 mM KCl, 0.34 µM biotin, and 0.75 µM thiamin). A comprehensive list of all nematode and fungal strains used in this study, along with their sources and genotypes, is available in S2 Table.

### Genetic screen, single-nucleotide polymorphism mapping, and whole-genome sequencing

Mutagenesis of *C. elegans* N2 was performed using EMS or ENU following established protocols to induce random mutations for phenotypic screening of trap resistance [39]. Semi-synchronized F2 populations were screened for fungal resistance by adding to *A. oligospora* cultures with pre-induced traps. One day before mutant screening, traps were induced on LNM plates by adding mixed stages of wild-type *C. elegans*. Mutants that resisted fungal trap capture were isolated, and their progeny underwent multiple rounds of re-screening to confirm the stability and reproducibility of the resistant phenotype. Approximately 200,000 genomes were screened, resulting in the isolation of ~30 mutants, of which five exhibited robust and reproducible resistance. These five mutants were selected for detailed genetic analysis, including SNP mapping and WGS.

For rapid SNP mapping, mutants were crossed with the Hawaiian strain (*C. elegans* CB4856) [11]. F2 progeny displaying trap resistance were selected for SNP mapping, and at least two individuals from each mutant line were analyzed for eight SNP markers per chromosome. Genomic DNA was extracted from these resistant F2 progeny and sequenced using Illumina WGS, with coverage depths ranging from 2 to 296× to ensure adequate resolution for mutation identification. The WGS data were processed and analyzed using the CloudMap pipeline, an open-source tool optimized for *C. elegans* mutation detection. Variant calling and SNP identification were performed using default parameters [40].

### Resistance assay

To quantify nematode trap resistance and fungal capture efficiency, we measured the nematode escape rate as a proxy. The laboratory strain *A. oligospora* TWF154 was cultured on 5-cm LNM plates at 25 °C for four days. Approximately

300–350 semi-synchronized N2 animals (L4 to young adult stages) were transferred onto the *A. oligospora* culture one day before the assay to stimulate trap formation. To assess trap resistance, 30 adult nematodes were transferred onto pre-induced fungal traps and allowed to crawl on the fungal lawn for 10 min. Following the incubation, ~10 ml of deionized water was gently applied to the plate to wash away untrapped nematodes. The nematode escape rate was calculated as the proportion of untrapped nematodes relative to the total number tested.

### Fosmid, collagen, and tissue-specific rescue

The WRM0634bB10 fosmid, containing the complete *nhr-66* locus, was injected into forward genetic screen (*yph406-408, 410,* and *412*) mutants and the *nhr-66* STOP-IN (*yph413*) mutant to restore *nhr-66* expression and assess functional rescue. The cDNA for *nhr-66* and other tested genes was amplified from N2 cDNA using Phusion High-Fidelity DNA Polymerase (Thermo Scientific) and subcloned into the pSM plasmid using the In-Fusion HD Cloning Kit (Takara Bio). Promoter swapping was achieved by subcloning various promoter sequences into the pSM plasmid using SphI, HindIII, and AscI restriction enzymes to drive tissue-specific expression of *nhr-66*. The resulting plasmids were microinjected into the gonads of the *nhr-66(yph413)* mutant.

Collagen genes were expressed under the control of a 2 kb *rol-6* promoter, selected for its strong and ubiquitous expression in the epidermis. These constructs were injected into the *nhr-66(yph413)* mutant.

### Reporter lines generation and fluorescence imaging

Plasmids of specific promoters driving GFP were generated by microinjection into either wild-type (N2) or *nhr-66(yph413)* mutant backgrounds. For the *nhr-66* GFP reporter, 2 kb of DNA upstream of the *nhr-66* isoform a was selected as the promoter.

Fluorescence imaging was performed using a Zeiss AxioImager Z1 microscope equipped with a CoolSNAP HQ2 camera. GFP expression was visualized under standardized imaging conditions, including a 40× objective and optimized exposure times to prevent saturation. Images were analyzed and processed using ZEISS ZEN Blue software. Briefly, images were quantified by manually framing individual worms in the images. The mean GFP fluorescence intensity within each framed nematode was calculated, and background brightness was subtracted

### Atomic force microscopy (AFM)

AFM imaging and quantification were performed following established protocols [41]. Young adult nematodes were paralyzed for imaging using 1 mM levamisole and immobilizing the head and tail using WORMGLU (GluStitch ) on a 3% agarose pad within a petri dish. To prevent dehydration, nematodes were immersed in 2.5 mL of M9 buffer throughout the imaging process.

AFM images were acquired using a JPK NanoWizard III atomic force microscope in contact mode for high-resolution surface imaging. Images were captured using qp-CONT-10 AFM probes (Nanosensors) with a set point of 0.35 V and a scanning speed of 1 Hz. The probe sensitivity and spring constant were calibrated using the JPK calibration tool to ensure measurement accuracy.

### Transmission electron microscopy (TEM)

Young adult *C. elegans* were prepared for TEM using high-pressure freezing for cryofixation. Nematodes were placed into a 100 μm-deep well of a type A planchette filled with *E. coli*, then covered with the flat side of a type B planchette to secure the sample. The assembled planchettes were immediately loaded into a freezing holder and cryofixed using a Leica HPM 100 high-pressure freezer. To prevent adhesion, planchettes were pre-coated with 1-hexadecene (Sigma-Aldrich, St. Louis, MO).

Following cryofixation, frozen planchettes were transferred to cryo-vials containing a freeze-substitution solution (0.1% uranyl acetate and 1% $OsO_4$ in acetone; Electron Microscopy Sciences, Hatfield, PA) and submerged in a liquid nitrogen bath. Freeze-substitution was performed using a Leica EM AFS2 system, following a controlled temperature regime that gradually transitioned from −90 °C to 20 °C to preserve ultrastructural integrity.

For resin infiltration, samples were placed in Spurr's resin/acetone (1:9) solution on a shaker for 48 hours, followed by stepwise resin concentration increases (20%, 50%, 75%, and 100%) at 8-h intervals. After the final infiltration in 100% resin, samples were embedded in fresh resin and polymerized at 70 °C for 12 h.

Ultrathin sections (75 nm) of the pharynx procorpus were collected on Formvar-coated EM grids (copper slot grids, 1 mm × 2 mm). Sections were post-stained with 1% aqueous uranyl acetate (UA) and Reynolds' lead citrate to enhance contrast. Imaging was performed using a Tecnai G2 Spirit TWIN (Thermo Fisher Scientific) transmission electron microscope operated at 100 kV, equipped with a GATAN CCD SC1000 camera (4,008 × 2,672 active pixels) for digital image acquisition.

Cuticle thickness and strut width were quantified using ImageJ. For cuticle thickness, five sites were selected per image, and their measurements were averaged to obtain a representative value for each image. The average values from images of the same nematode sample were then averaged to represent the overall cuticle thickness of the nematodes. For strut width quantification, the width was measured at the point of contact with the cortical or basal layer, depending which layer was in-frame. For images capturing the strut connected to both layers, either the cortical or basal side was chosen at random for quantification.

## Total RNA isolation, RNA-seq library preparation, and data analysis

**RNA isolation.** Late L4-stage *C. elegans* (*N2* and *nhr-66* STOP-IN mutant) were exposed to 9-cm, 5-day-old *A. oligospora* LNM cultures without traps for 30 min to study transcriptional responses under fungal interaction conditions. After exposure, nematodes were harvested and washed 2× in M9 buffer to remove residual bacteria and fungal debris, then flash-frozen in liquid nitrogen. Total RNA was extracted using a Trizol-based protocol [42], treated with Turbo DNase (Thermo Fisher Scientific) to remove genomic DNA, and further purified by ethanol precipitation. RNA quality was assessed using the Agilent Bioanalyzer (RNA Integrity Number, RIN ≥ 8.0), and concentrations were measured using a Qubit fluorometer (Thermo Fisher Scientific).

**RNA-seq library preparation.** RNA-seq libraries were prepared at the Genomics Core Facility, Institute of Molecular Biology, Academia Sinica, following standardized protocols [43]. Libraries were constructed using ribosomal RNA depletion and stranded RNA-seq methods to ensure accurate transcript quantification.

**Quality control and alignment.** Sequencing quality was evaluated using FastQC (v0.11.9) to confirm high-quality reads with minimal adapter contamination and sequence bias. Cleaned reads were aligned to the *C. elegans* reference genome (PRJNA13758, WormBase WS273) using STAR aligner (v2.7) [44] with default parameters. Gene- and isoform-level expression was quantified using RSEM (v1.3.3) [45].

**Differential gene expression and functional analysis.** Differential gene expression analysis was conducted using edgeR [46]. Raw counts were processed into a DGEList object, and low-abundance genes were filtered based on a count-per-million threshold across replicates. Gene expression was normalized using the trimmed mean of M-values method, and statistical significance was determined via exact tests, with false discovery rate adjustment. Dimension reduction for visualization was performed using PCA with the factoextra (v.1.0.7) package in R [47].

**Gene Ontology (GO) and visualization.** A total of 1,255 downregulated genes were subjected to GO enrichment analysis using WormBase Enrichment Suite [20]. Functional categories were considered significant with a q-value threshold of 0.1. Heat maps representing gene expression patterns were generated using the pheatmap (v.1.0.12) package in R. Differential Gene Expression analysis between N2 and *nhr-66* mutants files are available from the GEO database (GSM8967049-GSM8967054).

## Hypoosmotic stress assay

Hypoosmotic stress was assayed using a modified version of a previously described hypotonicity assay [48] according to suggestions from Nathalie Pujol (pers. communication). Synchronized day-1 adult nematodes were collected from NG plates using M9 buffer. The nematodes were washed twice with deionized water to remove residual bacteria and debris, followed by centrifugation at $500g$ for 30 s. The supernatant was carefully removed, and the nematode pellets were transferred to 24-well plates, with each well containing approximately 30–50 nematodes. The exact number of nematodes per well was recorded before adding 1 mL of deionized water to each well as the hypoosmotic medium. The number of intact nematodes was counted at specific time intervals—0, 5, 10, 15, 20, 25, 30, 45, 60, 90, 120, and 150 min. Nematodes displaying rupture of the cuticle or visible extrusion of internal contents were classified as burst and excluded from the intact count. The worm exploded rate was calculated as the percentage of burst nematodes relative to the initial count.

## Competition assay

Fifty milliliter centrifuge tubes were filled with autoclaved soil (Far East Flora Garden Online Jiffy Florafleur 002 Universal Potting Soil) to provide a controlled environment free from contaminants. Soil was autoclaved at 121 °C for 30 min and cooled before use. To each tube, 1 mL of *E. coli* OP50 culture, prepared as for NG plates (OD$_{600}$, 0.596), was added as a bacterial food source.

Synchronized GFP-labeled wild-type young adults (*C. elegans* strain SJ4144) and non-GFP *nhr-66(yph413)* mutants were washed twice in M9 buffer to remove residual bacteria and debris before being added to the tubes. Each tube received 1,000 GFP-labeled wild-type nematodes and 1,000 non-GFP *nhr-66(-)* nematodes. Tubes were loosely capped to allow gas exchange and incubated at 20 °C to simulate standard environmental conditions. At 7, 14, and 21 days, nematodes were harvested from the soil using the Baermann technique. The recovered nematodes were transferred to NG plates, and the numbers of GFP-labeled wild-type and non-GFP *nhr-66(-)* nematodes were quantified using a Zeiss steREO V20 microscope equipped with a Andor Zyla 5.5 sCMOS camera. GFP-positive nematodes were identified using a fluorescence filter (excitation 488 nm, emission 509 nm).

## Developmental stage assay

To quantify developmental stages, 10 day-1 adult hermaphrodites from both wild-type *C. elegans* (N2) and *nhr-66(yph413)* mutants were transferred to standard NG plates seeded with *E. coli* OP50. After two hours of egg-laying, the adult nematodes were removed by picking to prevent additional egg deposition. The plates were then incubated at 22 °C under standard laboratory conditions.

At 44, 56, 64, and 72 h post-egg lay, the developmental stages of the progeny were assessed under a dissecting microscope. Larval stages (L1, L2, L3, L4) and adults were identified based on size, gonad morphology, and overall appearance. The number of nematodes at each developmental stage was recorded for subsequent analysis.

## Brood size assay

To quantify brood size, L4-stage hermaphrodites from wild-type *C. elegans* (N2) and *nhr-66(yph413)* mutants were singled onto standard NG plates seeded with *E. coli* OP50. Each hermaphrodite was transferred daily to a fresh plate to prevent overcrowding and allow accurate progeny counts. Transfers continued until the mother ceased egg-laying. After progenies reached the adult stage, the total number of viable progenies produced by each hermaphrodite was manually counted under a dissecting microscope. Non-viable eggs or unfertilized oocytes were excluded from the count. The brood size for each nematode was recorded as the total number of hatched progenies.

## Dauer formation assay

The dauer formation assay was conducted as previously described [49]. Ten adult nematodes were transferred onto dauer induction plates and allowed to egg lay for three hours. Plates were incubated at 25 °C, a temperature known to enhance dauer induction under appropriate conditions. Adults were removed by picking, and the total number of eggs laid was counted under a dissecting microscope. Plates were then sealed with parafilm to prevent dehydration and stored at 25 °C for three days. After this period, nematodes were categorized as either adults or dauers based on morphology. Dauer formation rate was calculated as the proportion of dauers relative to the total number of nematodes observed.

## Mating efficiency assay

Mating efficiency was assessed using crosses between *dpy-13*(*e184*) mutant hermaphrodites and males from different strains. Three L4-stage *dpy-13(-)* hermaphrodites were placed on mating plates, which consisted of standard 5 cm NG agar plate seeded with *E. coli* OP50 make a 1 cm bacteria lawn. Eight L4-stage males were added to each mating plate, and the plates were maintained at 20 °C for 24 hours to allow mating.

After 24 h, the males were removed by picking, and the hermaphrodites were transferred daily to fresh NG plates until no additional progeny were produced. Once the progeny reached adulthood, offspring were categorized as dumpy (self-progeny) or non-dumpy (cross-progeny) based on morphological features such as body size and shape. Mating efficiency was calculated using the formula: mating efficiency = number of non-dumpy (cross progeny) divided by the total progeny.

## Supporting information

**S1 Data. This spreadsheet contains the data presented in the** Fig 1.
(XLSX)

**S2 Data. This spreadsheet contains the data presented in the** Fig 2.
(XLSX)

**S3 Data. This spreadsheet contains the data presented in the** Fig 3.
(XLSX)

**S4 Data. This spreadsheet contains the data presented in the** Fig 4.
(XLSX)

**S5 Data. This spreadsheet contains the data presented in the** Fig 5.
(XLSX)

**S6 Data. This spreadsheet contains the data presented in the S2–S6 Figs.**
(XLSX)

**S1 Table. CaeNDR-based annotation of *nhr-66* variants in wild-isolated *Caenorhabditis elegans* strains.**
(XLSX)

**S2 Table. Strains used in this study.**
(XLSX)

**S3 Table. Primers, Recombinant DNA, Chemicals, and Software used in this study.**
(XLSX)

**S1 Fig. Visualization of GFP driven by the *nhr-66* isoform a promoter in adult hermaphrodites.** White boxes indicate regions shown at higher magnification in subsequent panels (scale bar, 100 μm).
(TIF)

**S2 Fig. Developmental stages of wild-type *Caenorhabditis elegans* (N2) and *nhr-66 (yph413)* mutants at various time points post-oviposition.** (n is shown below the x-axis). The data underlying this figure can be found in S6 Data.
(TIF)

**S3 Fig. Dauer formation rate in wildtype and *nhr-66 (yph413)* mutants following treatment with extracted dauer pheromone.** (mean ± SEM, n is shown below the x-axis, two-tailed unpaired Student *t* test). The data underlying this figure can be found in S6 Data.
(TIF)

**S4 Fig. Brood size comparison between wildtype and *nhr-66 (yph413)* mutants.** (mean ± SEM, *n* is shown below the x-axis, two-tailed unpaired Student *t* test). The data underlying this figure can be found in S6 Data.
(TIF)

**S5 Fig. Mating efficiency of wildtype and *nhr-66 (yph413)* mutants at 20°C.** (mean ± SEM, *n* is shown below the x-axis, two-tailed unpaired Student *t* test). The data underlying this figure can be found in S6 Data.
(TIF)

**S6 Fig. Quantification of nematode escaping rates in wild-type *Caenorhabditis elegans* (N2), the *nhr-66 (yph413)* mutant, and *nhr-49 (yph475)* STOP-IN mutant.** (mean ± SEM; $n > 6$, two-tailed unpaired Student t *t*est). The data underlying this figure can be found in S6 Data.
(XLSX)

**S1 Video. Comparison of trapping interactions between A. oligospora and either N2 or forward genetic screen mutant(*yph406*).**
(MP4)

**S2 Video. Comparison of hypoosmotic stress tolerance between N2 and nhr-66 (yph413) mutants.**
(MP4)

## Acknowledgments

We thank the *Caenorhabditis* Genetic Center (CGC), the *Caenorhabditis* Natural Diversity Resource (CaeNDR), Dr. Erik Andersen, and Dr. Chun-Hao Chen for sharing the nematode strains, as well as Drs. Jonathan Ewbank and Nathalie Pujol for helpful suggestions on the hypoosmotic stress assay. We thank Sue-Ping Lee and Wen-Li Pon at the Imaging Core facility of Institute of Molecular Biology, Academia Sinica, for technical assistance with TEM. We also thank Chun-Liang Pan, Chun-Hao Chen, Erik Andersen, and Hillel Schwartz for comments and suggestions about this work. We acknowledge WormBase (https://doi.org/10.1093/genetics/iyae050) and CaeNDR.

## Author contributions

**Conceptualization:** Han-Wen Chang, Yen-Ping Hsueh.

**Formal analysis:** Han-Wen Chang, Hung-Che Lin.

**Funding acquisition:** Yen-Ping Hsueh.

**Investigation:** Han-Wen Chang, Hung-Che Lin, Ching-Ting Yang, Dao-Ming Chang.

**Methodology:** Han-Wen Chang, Yen-Ping Hsueh.

**Project administration:** Yen-Ping Hsueh.

**Resources:** Yi-Chung Tung, Yen-Ping Hsueh.

**Supervision:** Yen-Ping Hsueh.

**Validation:** Han-Wen Chang.

**Visualization:** Han-Wen Chang.

**Writing – original draft:** Han-Wen Chang.

**Writing – review & editing:** Rebecca J. Tay, Yi-Chung Tung, Yen-Ping Hsueh.

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
