## [Editor Report · Decision Letter 0]

Dear Ping, 

Thank you for submitting your manuscript entitled "Cuticular collagens mediate cross-kingdom predator-prey interactions between nematodes and trapping fungi" for consideration as a Research Article by PLOS Biology.

Your manuscript has now been evaluated by the PLOS Biology editorial staff, as well as by an academic editor with relevant expertise, and I'm writing to let you know that we would like to send your submission out for external peer review.

IMPORTANT: As previously mentioned, we think this would be best considered as an Update Article, and the Academic Editor agreed. No re-formatting is required, but please select "Update Article" as the article type when you upload your additional metadata (see next paragraph).

Once your full submission is complete, your paper will undergo a series of checks in preparation for peer review. After your manuscript has passed the checks it will be sent out for review. To provide the metadata for your submission, please Login to Editorial Manager (https://www.editorialmanager.com/pbiology) within two working days, i.e. by Apr 25 2025 11:59PM.

Kind regards,

Roli

Roland Roberts, PhD

Senior Editor

PLOS Biology

rroberts@plos.org

---

## [Decision Letter · Decision Letter 1]

Dear Ping,

Thank you for your patience while your manuscript "Cuticular collagens mediate cross-kingdom predator-prey interactions between nematodes and trapping fungi" was peer-reviewed at PLOS Biology. It has now been evaluated by the PLOS Biology editors, an Academic Editor with relevant expertise, and by three independent reviewers. 

You'll see that reviewer #1 is very positive, but presents a list of questions, some of which require textual or presentational changes, but some of which imply additional analyses. Reviewer #2 is also positive, but suggests a number of experiments (using a reporter line, attempting rescue with a different collagen gene). Reviewer #3 also praises the study; most of his/her requests are presentational, but points 4 and 5 suggest additional analysis and experiment, respectively.

Based on the reviews and on our Academic Editor's assessment, we are likely to accept this manuscript for publication, provided you satisfactorily address the points raised by the reviewers (by textual and/or presentational changes only) and the following data and other policy-related requests.

IMPORTANT - please attend to the following:

a) I discussed the reviewers' comments with the Academic Editor. While they recognise that some of the experimental and analytical requests would indeed improve the paper, they said that these are not required to support the current claims, and so we would only require textual and/or presentational changes to address the concerns raised.

b) Please address my Data Policy requests below; specifically, we need you to supply the numerical values underlying Figs 1BC, 2ABCD, 3ABCDFG, 4, 5BCD, S2, S3, S4, S5, either as a supplementary data file or as a permanent DOI’d deposition.

c) Please cite the location of the data clearly in all relevant main and supplementary Figure legends, e.g. “The data underlying this Figure can be found in S1 Data” or “The data underlying this Figure can be found in https://zenodo.org/records/XXXXXXXX

d) Please make any custom code available, either as a supplementary file or as part of your data deposition.

We expect to receive your revised manuscript within two weeks. 

*Published Peer Review History*

*Press*

Sincerely,

Roli

Roland Roberts, PhD

Senior Editor

rroberts@plos.org

PLOS Biology

DATA POLICY:

Regardless of the method selected, please ensure that you provide the individual numerical values that underlie the summary data displayed in the following figure panels as they are essential for readers to assess your analysis and to reproduce it: Figs 1BC, 2ABCD, 3ABCDFG, 4, 5BCD, S2, S3, S4, S5. NOTE: the numerical data provided should include all replicates AND the way in which the plotted mean and errors were derived (it should not present only the mean/average values).

CODE POLICY

DATA NOT SHOWN?

REVIEWERS' COMMENTS:

Reviewer #1:

Review of PLoS Biology

Organisms do not exist in isolation but in a complex ecological web with other organisms that involves predation, mutualism, and symbioses. Despite many examples of these highly nuanced and evolutionarily important relationships, little is known about the molecular basis underlying these interactions. One reason for this is because model systems - which have the genetic power to uncover molecular mechanisms - are often studied in lab settings outside their natural niches. One relationship that has captured much interest are worm-trapping fungi that capture comparatively large metazoans in predator-prey relationships. The present study is important because it exploits the powerful genetics of C. elegans to understand their vulnerabilities in this specific predator-prey relationship. A genetic screen was performed to look for predation-resistant mutants. This effort uncovered a transcription factor of the hormone receptor family. Further analysis revealed a new role of target adhesion proteins in predator escape, but not without a trade off in other physiologically relevant phenotypes. In addition to the uniqueness of the genetic screen, several other novel aspects of the paper include the ultrastructural analysis of the worm cuticle. In these ways, the study breaks new ground in an important and interesting area. Below follow some questions for the authors to consider: 

1. Why does the hormone receptor normally regulate collagen gene expression? Do worms have the option of altering their surface properties?

2. "Furthermore, sequence analysis of natural C. elegans populations revealed no obvious loss-of-function variants in nhr-66, suggesting selective pressures exist that balance adhesion-mediated predation risk with physiological robustness." Were changes observed in the collagen genes?

3. It is clear the authors are striving to make the study clear and relevant. Nevertheless, going from the results to Figure 1B and 1C is difficult for the general reader to understand. What's a fosmid? What is the purpose of the red rectangle? What are all the gene names? The figure legend does not clear this up.

4. For clarification, how many total collagen genes are there in worms?

5. Figure 2B, the wild type and mutant do not look the same - there is a difference in granularity. What do the authors think underlies this difference? 

6. Do the mutant worms have any motility defects? Do collagen overexpressing strains have altered motility?

7. Does purified collagen bind to the fungus?

8. Is cuticle left behind on the fungus after escape?

Reviewer #2:

This manuscript show very interesting results. To pick a few exciting points, they show that A. oligospora targets and kills the cuticle/collagen of the nematode; that nhr-66, which regulates the cuticle genes, was identified through forward genetics. 

They also show that nhr-66 is an essential gene selected for survival in nature by showing that there is no significant natural variations defective in nhr-66 functions. 

Overall, the manuscript will be appropriate for publication in PLOS Biology after a minor revision as suggested below. In addition, I felt that it would be interesting to examine natural variations of collagen genes controlled by nhr-66 to see if there is any selection during evolution.

Major points

- Figure 2: They describe that macroscopic cuticle structure has not been changed. Would it be possible to provide evidence for that, for example, by showing cuticle structures using translational reporter lines such as col-19::gfp in nhr-66 mutants?

- line 268: It would be nicer if they can show some other collagen genes that can rescue mutant phenotypes besides col-14.

- Other studies (for example, Nasrallah MA et al., 2023) seem to show that nhr-66 is a transcriptional repressor. It would be necessary to elaborate / explain their notion that nhr-66 is an activator.

Minor points

- line 148: Adding information on the insertion site STOP-IN cassette in Fig 1D would be helpful.

- line 163: Please revise the phrase 'the tissues key for cuticle synthesis' to 'the key tissues for cuticle synthesis'.

- line 194: It appears that only a single allele (yph413) was analyzed. Therefore, it would be better to revise the title from 'Mutations' to 'Mutation' to reflect this.

- Figure 2: Placing the Greek letters with the title of corresponding graphs would be easier for readers to comprehend.

- line 377: It is mentioned in discussion that nhr-49 mutant does not show trap resistance. However, relevant data are not present in the main text and figures. Please rephrase the sentence or show evidence.

Reviewer #3:

In this manuscript, the authors follow up on their prior study that identified a new family of fungal protein that mediate the trapping of prey nematodes. Using the fungus Arthrobotrys oligospora, they previously showed that the TEP genes encode proteins that facilitate the capture of C. elegans worms via a process that seemingly involved carbohydrate-mediated adhesion. Here, they employ a nifty genetic screen to identify C. elegans mutants that are capable of evading capture. Through detailed phenotypic characterization of the resulting nhr-66 mutants, the screen ultimately resulted in the demonstration that nematode collagens are likely adhesive targets that mediate capture by A. oligosperma. Although removing these targets confers resistance to capture, it comes with the associated cost of increased sensitivity to osmotic stress. Accordingly, this resistance mechanism is not observed in natural isolates. However, results from the study shed light on the initial adhesive interactions that mediate prey capture by A. oligosperma and identify strategies that could be used to engineer nematode-trapping fungi (NTF) as biocontrol agents in the field. 

Overall, this is a well-conceived and performed study that provides significant new insight into interactions between worms and NTF. The genetic, phenotypic, and molecular analysis are all first rate, such that the author's conclusions are sound and well-justified. Overall, I only have the following minor suggestions or concerns;

1. Fig. 1A would be much easier to interpret if the authors used a darker colour scheme to represent nematodes and the traps.

2. The plots presented in Figs. 2B and 2C are very difficult to read given the light colouring..

3. Line 288 and Fig 4A. In addition to A. musiformis, the protective affect of the nhr-66 mutation on A. amerospora is also reduced, although this occurs in a high background of general resistance to this species in wild type. This might be worth commenting on as well. 

4. Fig. 1B and Line 221. Was GO enrichment performed on the 168 up-regulated genes? Were any patterns observed?

5. Fig. 5B and Line 312. Were the natural variants within nhr-66 also tested for their effects on hypo-osmotoic stress? One would expect there to be a much lower frequency of ruptured worms given the negligible impact on escape.

---

## [Editor Report · Decision Letter 2]

Dear Ping,

Thank you for the submission of your revised Update Article "Cuticular collagens mediate cross-kingdom predator-prey interactions between trapping fungi and nematodes" for publication in PLOS Biology. On behalf of my colleagues and the Academic Editor, Aaron Mitchell, I'm pleased to say that we can in principle accept your manuscript for publication, provided you address any remaining formatting and reporting issues. These will be detailed in an email you should receive within 2-3 business days from our colleagues in the journal operations team; no action is required from you until then. Please note that we will not be able to formally accept your manuscript and schedule it for publication until you have completed any requested changes.

Sincerely, 

Roli

Senior Editor

PLOS Biology

rroberts@plos.org